# S-Flow GAN

## Abstract

Our work offers a new method for domain translation from semantic label maps and Computer Graphic (CG) simulation edge map images to photo-realistic images. We train a Generative Adversarial Network (GAN) in a conditional way to generate a photo-realistic version of a given CG scene. Existing architectures of GANs still lack the photo-realism capabilities needed to train DNNs for computer vision tasks, we address this issue by embedding edge maps, and training it in an adversarial mode 1. We also offer an extension to our model that uses our GAN architecture to create visually appealing and temporally coherent videos.

## 1 Introduction

The topic of image to image translation and more generally video to video translation is of major importance for training autonomous systems. It is beneficial to train an autonomous agent in real environments, but not practical, since enough data cannot be gathered Collins et al. (2018). However, using simulated scenes for training might lack details since a synthetic image will not be photo-realistic and will lack the variability and randomness of real images, causing training to succeed up to a certain point. This gap is also referred to as the reality gap Collins et al. (2018). By combining a non photo-realistic, simulated model with an available dataset, we can generate diverse scenes containing numerous types of objects, lightning conditions, colorization etc. Chen & Koltun (2017).

In this paper, we depict a new approach to generate images from a semantic label map and a flexible Deep Convolution Neural Network (DCNN) we called Deep Neural Edge Detector (DNED) which embed edge maps. we combine embedded edge maps which act as a skeleton with a semantic map as input to our model (fig 2.1), The model outputs a photo-realistic version of that scene. Using the skeleton by itself will generate images that lack variability as it restricts the representation to that specific skeleton itself. Instead, we learn to represent skeletons by a neural network and at test time, we sample the closest appropriate skeleton the network has seen at training. Moreover, we have extended this idea to generate photo-realistic videos (i.e. sequence of images) with a novel loss that uses the optical flow algorithm for pixel coherency between consecutive images.

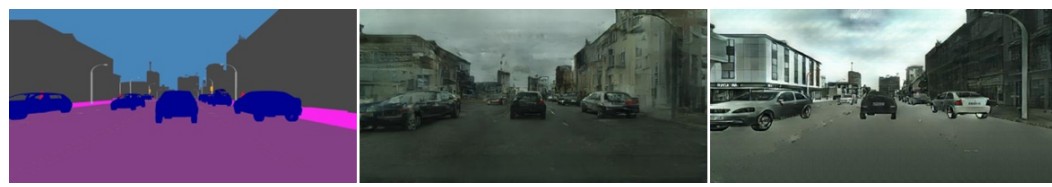

Figure 1: in this paper we propose a method for generating photo-realistic images from semantic labels of a simulator scene. This figure provides images related to the Synthia dataset Ros et al. (2016). Left - semantic map of the scene. Middle - generated image from pix2pixHD Wang et al. (2018b). Right - Our generated image. The texture and color space in our generated image is more natural giving the image the desired photo-realism.

Recent works in the field of image generation include pix2pix Isola et al. (2017) offering image generation from semantic maps, cascaded refinement networks Chen & Koltun (2017) using networks refining different resolutions in a cascade manner, pix2pixHD Wang et al. (2018b) can generate HD images in a conditional manner using multi-scale discriminator and an dual generator

used as a super resolution generator. L1 loss for image generation is known to generate low quality images as the generated images are blurred and lack details Dosovitskiy & Brox (2016). Instead, Gatys et al. (2016), Johnson et al. (2016) are using a modified version of the perceptual loss, allowing generation of finer details in an image. Pix2pixHD Wang et al. (2018b) and CRN Chen & Koltun (2017) are using a perceptual loss as well for training their networks, e.g. VGGnet Simonyan & Zisserman (2014). Moreover, pix2pixHD are using instance maps as well as label maps to enable the generator to separate several objects of the same semantics. This is of high importance when synthesizing images having many instances of the same semantics in a single frame.

As for video generation the loss used by Wang et al. (2018a), Shahar et al. (2011) tend to be computationally expensive while our approach is simpler. We are using two generators of the same architecture, and they are mutually trained using our new optical flow based loss that is fed by dense optical flow estimation. Our evaluation method is FID Heusel et al. (2017) and FVD Unterthiner et al. (2018) as it is a common metric being used for image and video generation schemes. We call this work s-Flow GAN since we embed Spatial information obtained from dense optical flow in a neural network as a prior for image generation and flow maps for video coherency. This optical flow is available since the simulated image is accessible at test time in the case of CG2real scheme.

We make Three major contributions: First, our model can generate visually appealing photo-realistic images from semantic maps having high definition details. Second, we incorporate a neural network to embed edge maps, thus allowing generation of diverse versions of the same scenes. Third, we offer a new loss function for generating natural looking videos using the above mentioned image generation scheme. please refer to this link for videos and comparison to related work.

## 2 RELATED WORK

### 2.1 GENERATIVE ADVERSARIAL NETWORKS

Generative Adversarial Networks (GAN) were introduced in 2014 Goodfellow et al. (2014). This method generate images that look authentic to human observers. They do so by having two neural networks, one generating candidates while the other acts as a critique and tries to evaluate the generation quality Arjovsky et al. (2017),Radford et al. (2015),Zhao et al. (2016),Zhu et al. (2016),Salimans et al. (2016). GANs are widely used for image generation; some image synthesis schemes are used to generate low resolution images e.g. 32x32 Isola et al. (2017) while Brock et al. (2018) were able to generate higher resolution images (up to 512x512). In addition, Wang et al. (2018b) were able to generate even higher resolution images using coarse-to-fine generators. The reason generating high resolution images is challenging is the high dimensionality of the image generation task and the need to provide queues for high resolution Odena et al. (2017), Karras et al. (2017). We offer queues as an edge map skeletons generated by our proposed DNED module. During training the DNED is trained to learn the representations of real image edge maps. During test the DNED is shown a CG (Computer Graphics) edge map, finds its best representation and provides the generator with an appropriate generated edge map sampled from real image edge maps distribution.

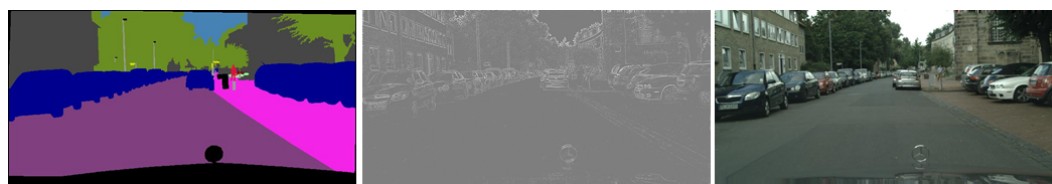

Figure 2: example of the images for training the model. Left is the semantic map. Middle is the edge map extracted from the real image. Right is the real image being used by the discriminator for adversarial training. Please note that the real image is not used by the generator neither at training nor at test time, but only its edge map. The main issue in the CG2real model compared to the image to image models is that the simulators image is available to the generator at test time. We thus use the simulators image to extract the edge map allowing the generator to generate the necessary fine details in the output image.

## 2.2 IMAGE SYNTHESIS

### 2.2.1 IMAGE TO IMAGE TRANSLATION

In the pix2pix setting, they used a Conditional GAN Mirza & Osindero (2014), where the networks input is a semantic map of the scene, and while training in adversarial mode, a fake version of the real image is given to the discriminator to distinguish. In the CG2real setting in addition to the semantic map we also have access to the simulated image. Using the CG image as is, might be counter productive since it will be trained to reconstruct CG images and not photo-realistic ones. Conversely some of the underlying CG information correlates with the real world and can provide meaningful prior to the synthesis. Since the relevant information lies in the image high frequencies Burt & Adelson (1983), we learn the distribution of edge maps in real images (high resolution details), and provide representation of it to the generator at test time. Some image generation tasks use label maps only, e.g. Isola et al. (2017). The label maps provide only information about the class of a given pixel. In order to generate photo-realistic images, some use instance maps as well Wang et al. (2018b), This way, they can differentiate several adjacent objects of the same class. Nonetheless, while most datasets provide object level information about classes like cars, pedestrians, etc. they do not provide that information about vegetation and buildings. As a result, the generated images might not correctly separate those adjacent objects, thus degrading photo-realism.

### 2.2.2 LEARNING EDGES BY A NEURAL NETWORK

Generating edge maps using neural networks is a well established method. Holistically-Nested Edge Detection (HED) provides holistic image training and prediction for multi-scale and multi-level feature learning Xie & Tu (2017). They use a composition of generated edge maps to learn a fine description of the edge scene. Inspired by their work, we train a neural network to learn edge maps of real images.

As mentioned before, our generator requires an edge map as input. we get the edge map using a spacial Laplacian operator with threshold. Providing the generator with deterministic edge map will produce the same scene, so we train the DNED to take as input that deterministic edge map, learn its representation and produce a variant of that edge map, as a superposition of edges seen in real datasets. This way the generator will be able to produce a varaiaty of photorealistic images for the same scene.

Since our approach (using edge maps) is not class dependent, we do not need instance map information to generate several adjacent instances of the same semantics. Moreover, this approach addresses the problem of generating fine details within a class like buildings and vegetation as can bee seen in fig 3.2.

## 2.3 VIDEO TO VIDEO SYNTHESIS

Generating temporally coherent image sequences is a known challenge. Recent works use GANs to generate videos in an unconditional setting Saito et al. (2017),Tulyakov et al. (2018),Vondrick et al. (2016), by sampling from a random vector, but don't provide the generator with temporal constrains, thus generating non coherent sequences of images. Other works like video matting Bai et al. (2009) and video inpainting Wexler et al. (2004) translate videos to videos but rely on problem specific constrains and designs. A recent work named vid2vid Wang et al. (2018a) offers to conditionally generate video from video and is considered to one of the best approaches to date. Using FlowNet 2.0 Ilg et al. (2017) they predict the optical flow of the next image. In addition, they use a mask to differentiate between two parts; the hallucinated image generated from instance-level semantic segmentation masks and the predicted image from the previous frame. By adding these two parts, this method can combine the predicted details from the previously generated image, with the details from the newly generated image. Inspired by Wang et al. (2018a), we are using flow maps of consecutive images to generate temporally coherent videos. Contrary to Wang et al. (2018a) we are not using a CNN to predict the flow maps or a sequence generator, but a classical Computer vision approach. This is since a pre-trained network (trained on real datasets) failed to generalize and infer on simulated datasets e.g. Synthia. This enables better temporal coherency and improve video generation robustness.

# 3 MODEL

Our CG2real model aims to learn the conditional distribution of an image given a semantic map. Our video generation model aims to use this learned distribution for generating temporally coherent videos using the generated images from the CG2real scheme. We first depict the image generation scheme, then we review our video generation model.

## 3.1 IMAGE GENERATION

We use a conditional GAN to generate images from semantic maps as in Isola et al. (2017). In order to generate images, the generator receives the semantic segmentation images $s_i$ and maps it to photo-realistic images $x_i$. In parallel, the discriminator takes two images, The real image $x_i$ (ground truth) and the generated image $f_i$ and learns to distinguish between them. This supervised learning scheme is trained in the well-known min max game Goodfellow et al. (2014),Salimans et al. (2016):

$$\min_G \ \max_D \mathcal{L}_{GAN}(D, G) \tag{1}$$

$$\mathcal{L}_{GAN(D,G)} = E_{(x,s)}[log(x, s)] + E_{(s \sim p_{data}(s))}[log(1 - D(s, G(s)))] \tag{2}$$

## 3.2 EMBEDDING EDGE MAPS

In order to generate photo-realistic visually appealing images containing fine details, we provide a learnt representation of an edge map to the generator (fig 2.1), allowing it to learn the conditional distribution of real images given semantic maps and edge maps, i.e.:

$$\mathcal{L}_{GAN(D,G,e)} = E_{(x,s)}[log(x, s)] + E_{((s,e) \sim p_{data}(s,e))}[log(1 - D(s, G(s, e)))] \tag{3}$$

During training, given an example image $x_i$, we can estimate its edge map by the well-known spatial Laplacian operator Guattery & Miller (2000),Denton et al. (2015). This edge map is concatenated to the semantic label map and both are given as priors to the generator for adversarial training of the fake image $f_i$ vs. the real image $x_i$. To allow a stable training we begin training our GAN with the edge maps from the Laplacian operator. After stabilization of the generator and discriminator, we provide our generator with edge maps from the DNED. We then jointly train the GAN with the DNED.

The DNED architecture is a modified version of HED Xie & Tu (2017). In HED, they generate several sized versions of the edge map, each having a different receptive field. The purpose is to create an ensemble of edge maps, each allowing different level of details in the image. When superimposing all, the resulting edge map will have coarse-to-fine level of details in the generated edge map image. By changing the weights of that ensemble, we can generate the desired variability in the generated edge map, thus allowing us to generate diverse versions of the output. To conclude, the loss function for training the DNED is:

$$\mathcal{L}_{DNED} := \mathcal{L}_{DNED}(E(x)) = \sum_{i=1}^{N} a_i * BCE(d_i(x), E(x)) \tag{4}$$

Where: $d_i(x)$, $i = 0 : 5$ is the $i^{th}$ side output of a single scale, $E(x)$ is the classic edge map generated by the spatial Laplacian operator, BCE is the binary cross entropy loss. $N = 6$ in our case. $a_i$ is the contribution of the $i^{th}$ scale to the ensemble.

Increasing the resolution of the image might be challenging for GAN training. In other methods the discriminator needs a large receptive field Isola et al. (2017),Seif & Androutsos (2018),Simonyan & Zisserman (2014),Luo et al. (2016), requiring a deeper network or larger convolution kernels. Using a deeper network is prone to overfitting and in the case of GAN training, and might cause training to be unstably. This challenge is usually addressed by the multi-scale approach Ghiasi & Fowlkes (2016),Denton et al. (2015),Huang et al. (2017),Karras et al. (2017),Zhang et al. (2017).

Since the DNED embed a learnt representation of skeletons, our architecture performs very well on higher resolution images. Our original generated images were of size [512x256]. We have successfully trained our model to generate images of size [768x384] , i.e. 1.5 times larger in each dimension without changing the model while using a single discriminator (see 3.2).

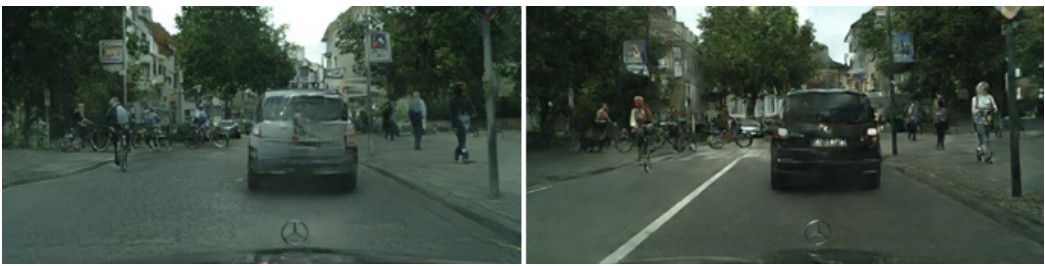

Figure 3: comparison of 768x384 pix images generated by pix2pixHD (Left) and our model (Right). Our model can generate lower level details in the image, thus improving its photo-realism. This figure provides an example comparing (768x384 pix) resolution images of pix2pixHD (left side) compared to our model (right).

We showed that generating high quality images when using a single discriminator is feasible and training is stable. We provide comparison using our method with multi-scale discriminator 3.2. the FM loss is computed with k=1 for single layer discriminator and k=3 for multi layer one:

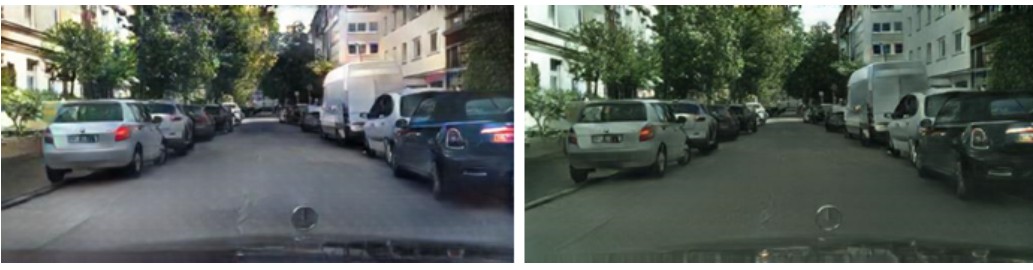

Figure 4: Comparison of generated test images, when training with a single discriminator and a multi-scale one. The left image is generated when the generator was trained with a single discriminator, while the right image while using a multi-scale one. This figure demonstrates that when using our model (with our skeleton), training with a single discriminator might be enough.

$$\mathcal{L}_{FM_m}^k : = \mathcal{L}_{FM_m}^k(D_k, G, e) = \sum_{i=1}^{T} \frac{1}{N_i} E_{(x,s,e)\sim p_{data}(s,x,e)} \mathcal{L}_1(D_k^i(s, x) - D_k^i(s, G(s, e))) \quad (5)$$

In addition, following Dosovitskiy & Brox (2016),Gatys et al. (2016),Johnson et al. (2016),Zhu et al. (2017) we are using the perceptual loss for improved visual performance and to encourage the discriminator distinguish real or fake samples using a pre traind VGGnet Ledig et al. (2017).

$$\mathcal{L}_{percep} : = \mathcal{L}_{percep}(x, G(s, e)) = \frac{1}{P} \sum_{i=1}^{P} \mathcal{L}_1(FL_{VGG_i}(x) - FL_{VGG_i}(G(s, e))) \quad (6)$$

Where, P is the number of slices from a pre-trained VGG network and $FL_{VGG_i}$ are the features extracted by the VGG network from the $i^{th}$ layer of the real and generated images respectively. To conclude, our overall objective for generating photo-realistic, diverse images in the CG2real setting is to minimize $L_{CG2real}$:

$$L_{CG2real} = \min_{G} \max_{D_k, k=1:l_m} \sum_{l=1}^{l_m} \mathcal{L}_{GAN}(D_k, G, e) + \lambda_1 \sum_{l=1}^{l_m} \mathcal{L}_{FM_m}^k + \lambda_2 \mathcal{L}_{percep} + \lambda_3 \mathcal{L}_{NNED} \quad (7)$$

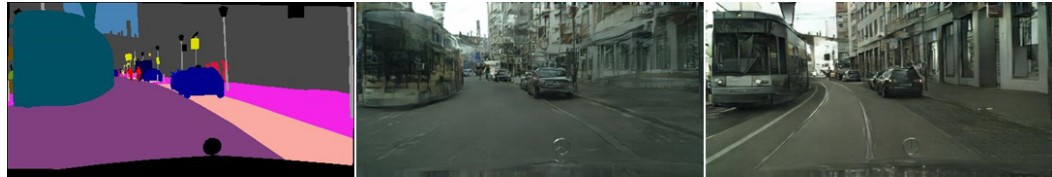

Figure 5: By using edge maps, the model learns to separate objects of the same semantics. The most dominant example is buildings. Unlike cars, pedestrians or bicycle riders, that are separable using the instance map, buildings are not. The semantic label provides the pixels in which the building exists. Considering the fact that a scene of adjacent buildings is somewhat common, the ability to separate them is of high value. Left - the label map. Middle - generated image by Wang et al. (2018b). Right - our generated image. Our model can generate unique adjacent buildings from the semantic label maps of better quality compared to Wang et al. (2018b).

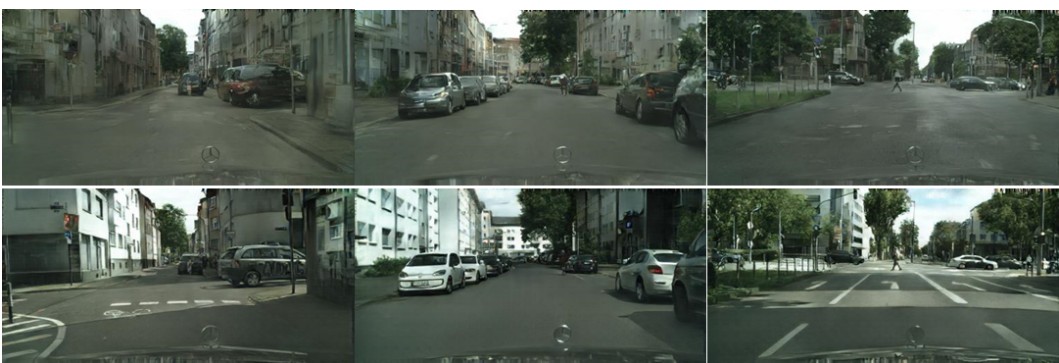

Figure 6: previous work test images Wang et al. (2018b) (Top) compared to our model test images (Bottom). The images generated by our model contain low level details, allowing the desired photorealism

## 3.3 VIDEO GENERATION

Using pre trained CG2real networks, we generate two consecutive images, and then estimate two flow maps. The first flow map is between $x_i, x_{i+1}$, where $x_i$ and $x_{i+1}$ are two consecutive real images. The second flow map is between $G(s_i, e_i), G(s_{i+1}, e_{i+1})$, where $G(s_i, e_i)$ and $G(s_{i+1}, e_{i+1})$ are two consecutive generated (fake) images. Note that the generation of $G(s_i, e_i), G(s_{i+1}, e_{i+1})$ is done independently, meaning we apply our CG2real method twice, without any modifications. To conclude we enforce temporal coherency by using the following loss:

$$\mathcal{L}_{flow} = \mathcal{L}_1(\mathcal{F}_{real}, \mathcal{F}_{fake}) \tag{8}$$

Where $\mathcal{F}_{real} = \mathcal{F}(x_i, x_{i+1})$, $\mathcal{F}_{fake} = \mathcal{F}(G(s_i, e_i), G(s_{i+1}, e_{i+1}))$ and $\mathcal{F}(*)$ is the optical flow operator. This formulation eliminates the need of using a sequential generator as in Wang et al. (2018a), allowing us not only using our image generation model twice, which adds more constrains to the video generation scheme, but also avoid errors accumulation arising from positive feedback by feeding a generated image to the generator, as can be seen in figure 3.3 and in this video.

By adding $L_{flow}$ to the $L_{CG2real}$ loss, the network learns to generate $G(s_{i+1}, e_{i+1})$ taking the flow maps into account, thus generating temporally coherent images as depicted in 3.3.

$$\mathcal{L}_{videogen} = \mathcal{L}_{flow} + \mathcal{L}_{CG2real} \tag{9}$$

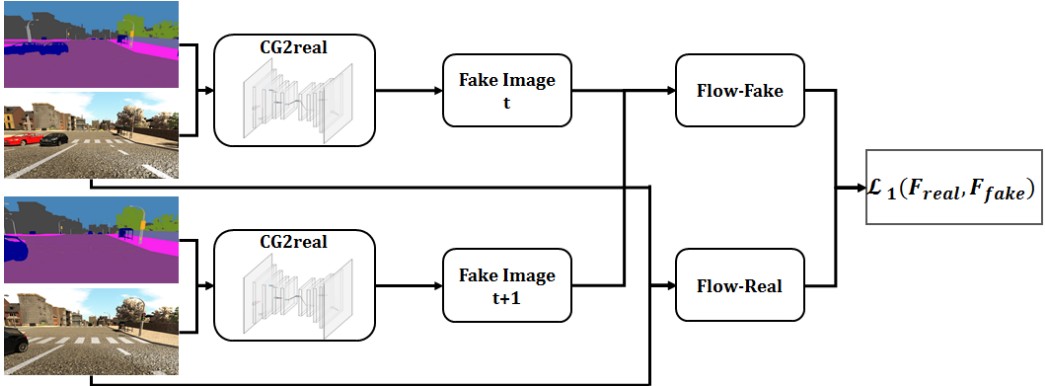

Figure 7: block diagram of the video generation model. Two identical CG2real models generate Fake image (t) and Fake image (t+1). The two consecutive fake images are fed to the flow-fake estimator, while two consecutive real images are fed to the flow-real estimator. Both real and fake flow maps are trained using $L_1(F_{real}, F_{fake})$ loss. This enables the pre-trained CG2real models to learn the required coherency for generating photo-realistic videos.

## 4 RESULTS

Our goal is to generate photo-realistic images. In (fig 3.2) we can find some examples from the CG2real image synthesis task, and in (fig 4) present consecutive images depicting the video to video synthesis. We use the same evaluation methods as used by previous image to image works ,e.g. pix2pix Isola et al. (2017) , pix2pixHD Wang et al. (2018b) and others. The evaluation process consist of performing semantic segmentation with a pre-trained seamntic segmentation network Zhao et al. (2017) on synthesized images produces by our model, then calculating the semantic pixel accuracy and the mean intersection over union (mIoU) over the classes in the dataset. As shown in tables 1, 2 bellow, our network outperforms previous works. The ground-truth results are the pixel accuracy and mIoU when performing the same semantic segmentation with the real images (Oracle).

Furthermore, to evaluate the image generation quality, we used another metric to evaluate distances between datasets called FID (Frchet Inception Distance) Heusel et al. (2017),Adler & Lunz (2018). It is a very common metric for generative models as it correlates well with the visual quality of generated samples Wang et al. (2018a). FID calculates the distance between two multivariate Gaussians real and generated respectively; where $X_r \sim N(\mu_r, \Sigma_r)$ and $X_g \sim N(\mu_g, \Sigma_g)$ are the 2048-dimensional activations of the Inception-v3 pool3 layer Szegedy et al. (2016), and $FID = \|\mu_r - \mu_g\|^2 + Tr(\Sigma_r + \Sigma_g - 2(\Sigma_r \Sigma_g)^{1/2})$ is the score for image distributions $X_r$ and $X_g$. Lower FID score is better, meaning higher similarity between real and generated samples.

| Cityscapes | Pix2pix | Pix2pixHD | Ours | Oracle |
|---|---|---|---|---|
| Pixel accuracy [%] | 0.7279 | 0.81 | 0.83 | 0.86 |
| Mean IoU [%] | 0.5324 | 0.67 | 0.69 | 0.701 |

Table 1: semantic segmentation results on the cityscapes Cordts et al. (2016) validation set

| Synthia | Pix2pix | Pix2pixHD | Ours | Oracle |
|---|---|---|---|---|
| Pixel accuracy [%] | 0.54 | 0.79944 | 0.860753 | 0.913132 |
| Mean IoU [%] | 0.36 | 0.55955 | 0.740040 | 0.8419 |

Table 2: semantic segmentation results on the Synthia Ros et al. (2016) dataset

As can be seen in tables 1, 2, pix2pixHDs results are better than pix2pix for pixel accuracy and mIoU. Our results are better than pix2pixHD, and almost meet the oracles results on both Synthia Ros et al. (2016) and cityscapes Cordts et al. (2016). In table 3, we compare the FID score for all

| FID,FVD | Pix2pix | Pix2pixHD | Vid2vid | Ours-img | Ours-vid |
|---------|---------|-----------|---------|----------|----------|
| FID     | 116.69  | 71.21     | 154.36  | 69.25    | 69.81    |
| FVD     | -       | -         | 0.706   | -        | 0.326    |

Table 3: FID and FVD metric comparisson between pix2pix, pix2pixHD vid2vid and Ours.

the four image generation models w.r.t the Oracle. Ours-img (Our image generation model) outperforms both pix2pix and pix2pixHD. Moreover, adding a temporal consistency constrain to the image generation process degrades image quality. Vid2vid uses pix2pixHD as its image generation model imposes a substantial degradation in the image quality (71.21 to 154.36). Our video generation uses our CG2real model had a marginal effect on the FID score of Ours-vid (69.25 to 69.81 and even outperformed pix2pixHD) and did not degrade generated images quality (fig 4).

Our video generation evaluation method is FVD (Frchet Video Distance) proposed by Unterthiner et al. (2018). FVD is a metric for video generation models evaluation and it uses a modified version of FID. we calculated the FVD score for our generated video (Ours-vid) w.r.t. the Oracle (real video) and did the same for vid2vid w.r.t the same Oracle. Our FVD score on the video test set is 0.326 while vid2vid's is 0.706 meaning our videos are more than twice similar to the oracle. we suggest that this substantial margin stems from the errors accumulated in the video generation model of vid2vid (fig 4). As mentioned, Our video generation model uses our flow loss therefore does not encounter this phenomena.

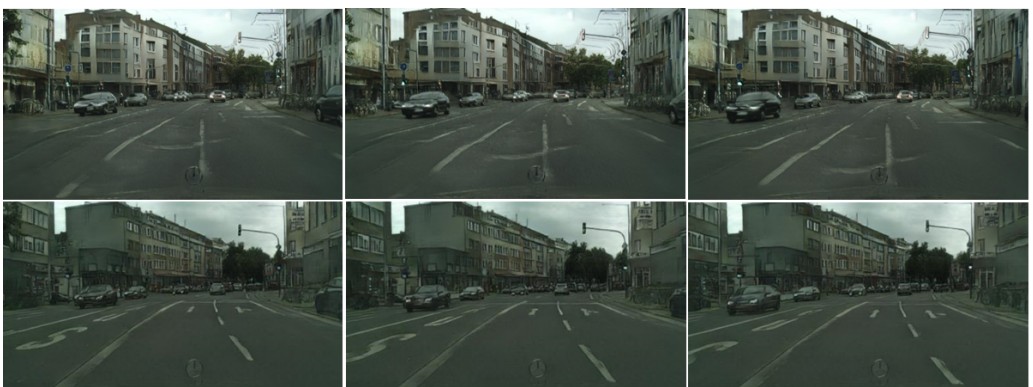

Figure 8: Comparison of video generation. Up - images generated by vid2vid Wang et al. (2018a). Down - images generated by our video generation model. Our generated images are temporally coherent and visually appealing In our images sky is more natural, road signs are clearer and buildings have finer level of details. This example emphasizes the error propagation of vid2vid's model wile our model does not accumulate errors (see street lights in upper right corner of each image). The main objective of the video generation model is to enable generating non flickering images by giving objects in consecutive images the same color and texture, i.e. sample from the same area in the latent spaces. full videos can be seen here .

## 5 SUMMARY

We present a CG2real conditional image generation as well as a conditional video synthesis. We offer to use a network learning the distribution of edge maps from real images and integrate it into a generator (DNED). We were able to generate highly detailed and diverse images thus enabling better photo-realism. Using the DNED enable generating diverse yet photo-realistic realizations of the same desired scene without using instance maps. As for video generation, we offer a new scheme that utilizes flow maps allowing better temporal coherence in videos. We compared our model to recent works and found that it outperforms both current quantitative results and more importantly generates appealing images. Furthermore, our video generation model generates temporally coherent and consistent videos.

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

# A  APPENDIX

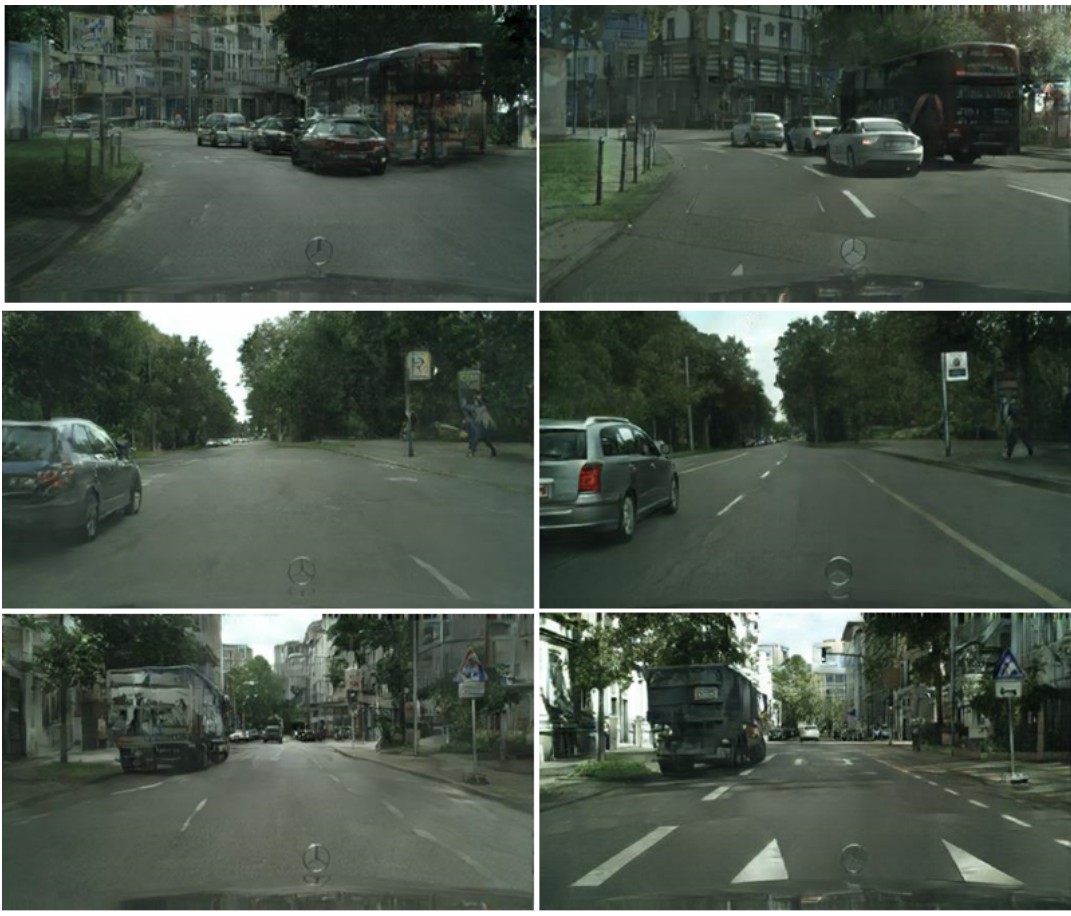

Figure 9: Additional Test images from Cityscapes Dataset. Left - pix2pixHD. Right - Ours. These images further demonstrate the photo-realism achieved by our model.

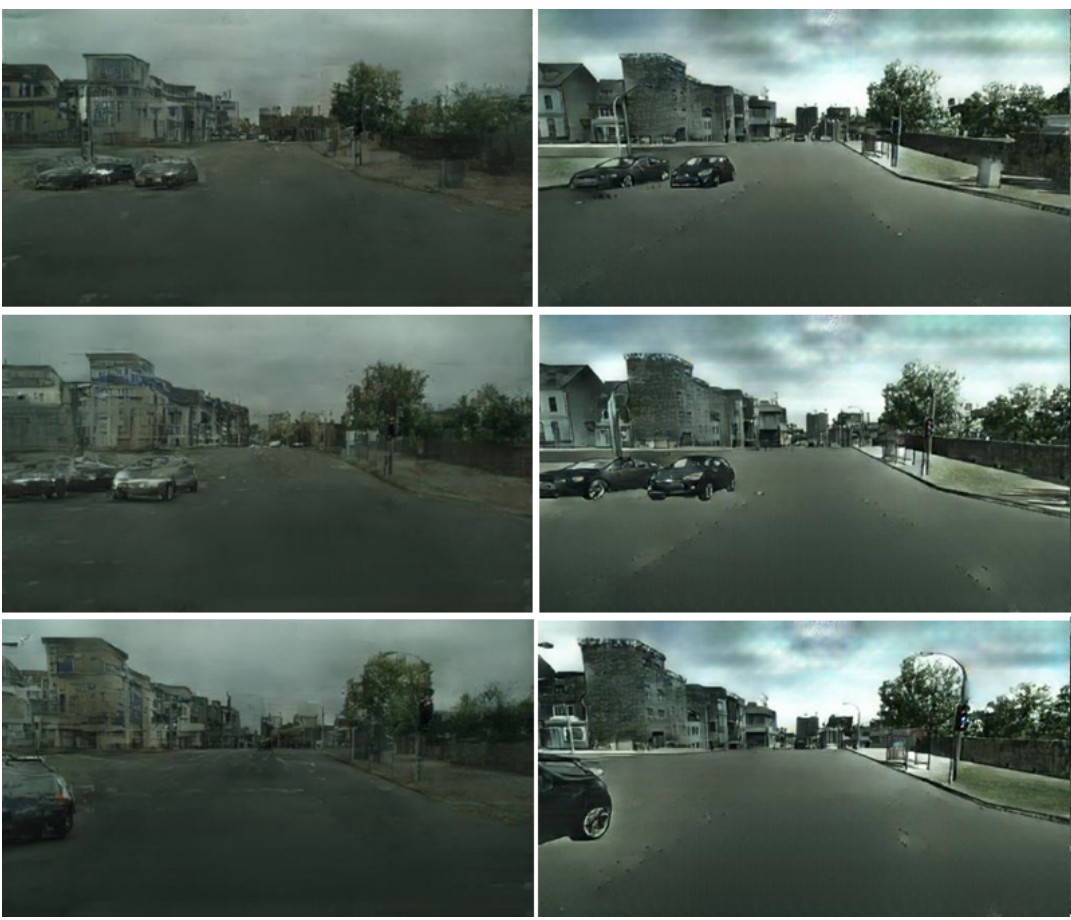

Figure 10: Additional test images on Synthia dataset. Left - pix2pixHD. Right - Ours. These images demonstrate improved image quality, better and finer details in the generated objects, buildings and vegetation.

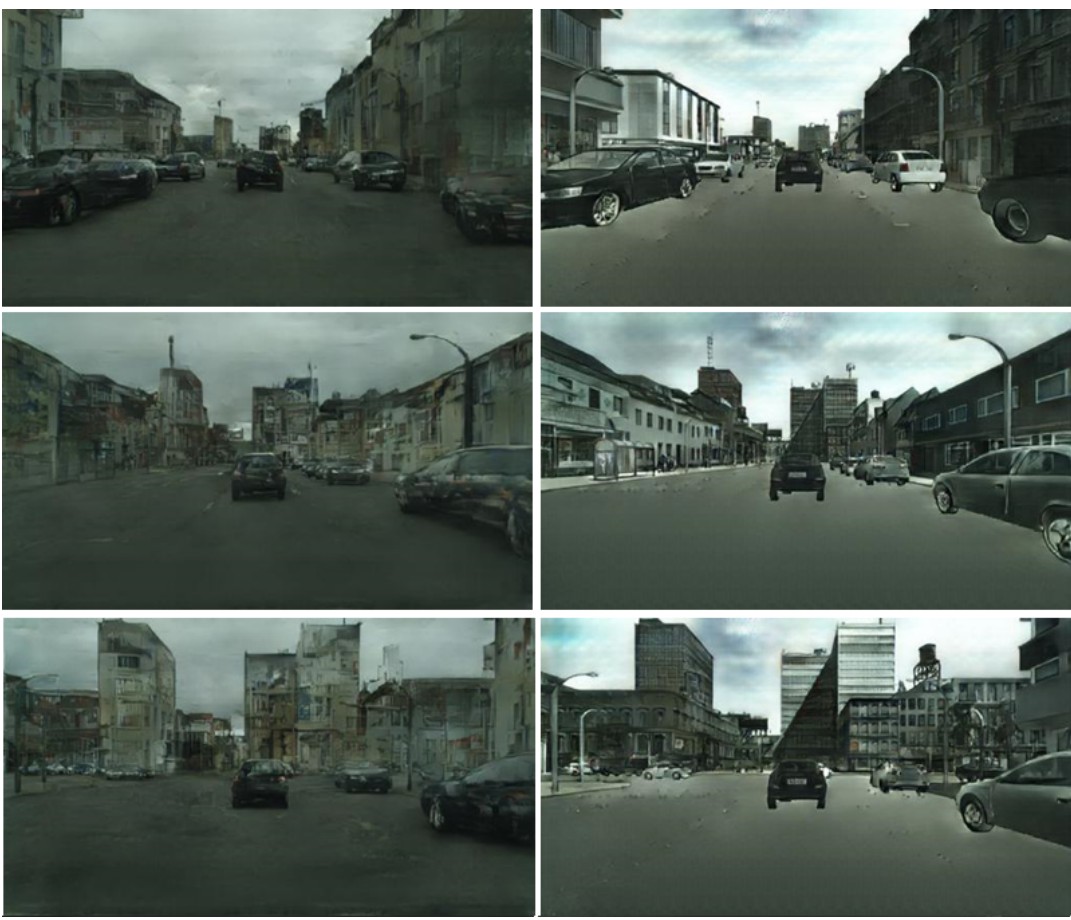

Figure 11: Additional test images on Synthia dataset. Left - pix2pixHD. Right - Ours. These images demonstrate improved image quality, better and finer details in the generated objects, buildings and vegetation.

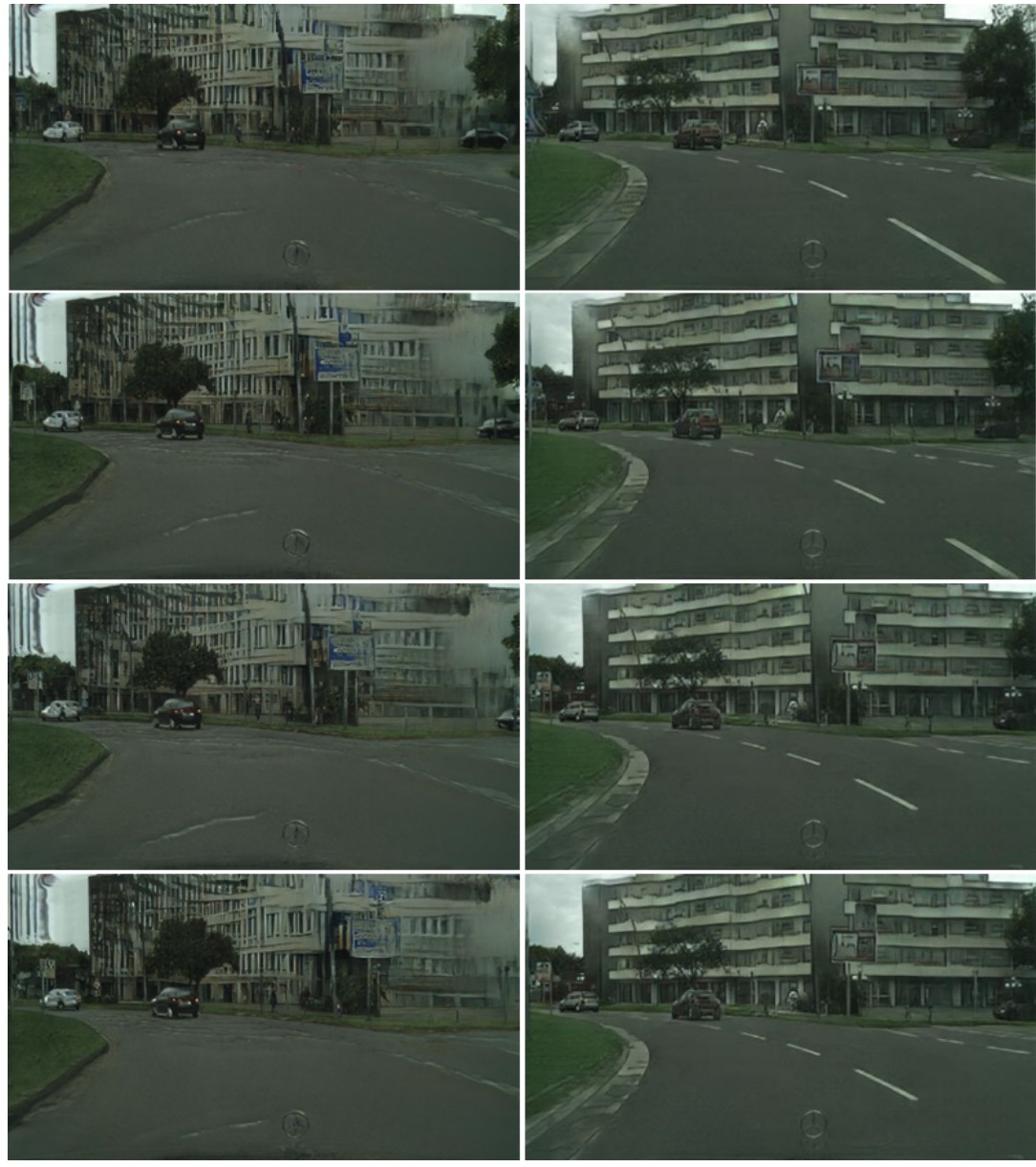

Figure 12: Test video on CityScapes. Left - vid2vid. Right - Ours video gen model. These images demonstrate better temporal coherency in the generated images. Moreover, in the top left corner of the left video, we notice the error propagates. Better yet, the buildings in the right video are more reasonable, w.r.t. windows, shades, general texture, etc. As for image quality, the road signs in the right video are better emphasized.

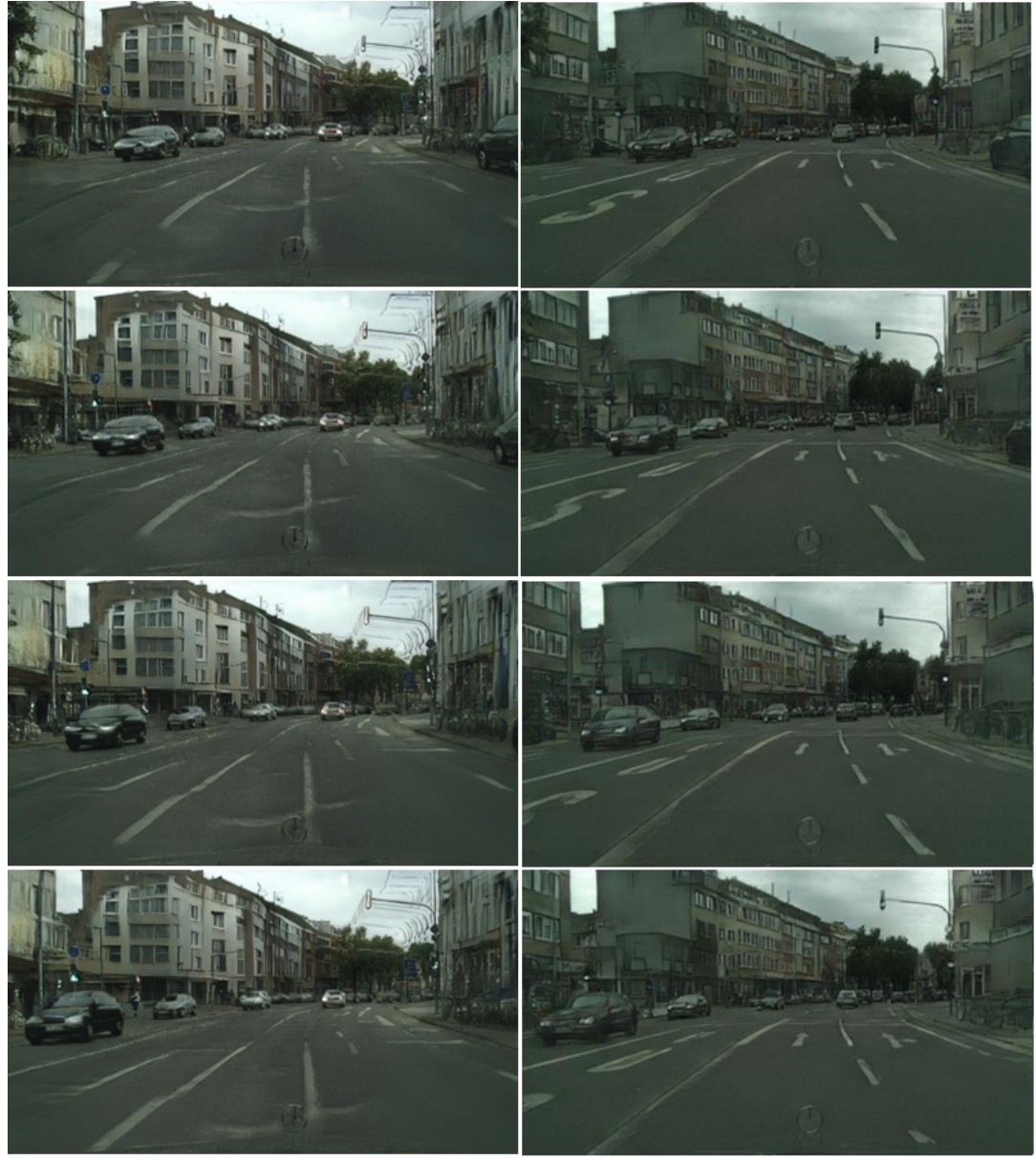

Figure 13: Test video on CityScapes. Left - vid2vid. Right - Ours video gen model. This figure provides more images from the same video presented in fig 4. Pay attention to the error propagation on the top right images of vid2vid. Again, our model demonstrates finer road signs and higher level of details in the generated buildings.

