# OpenReview forum: "S-Flow GAN"
_ICLR.cc/2020/Conference — Reject_

### Official Review · AnonReviewer1 · 2019-10-22
**Official Blind Review #1**

**Rating:** 1

**Review:**

This paper proposed a method to improve the image-to-image translation. By utilizing the CG-based Synthia and embedded edge maps, it has shown some effects on the Cityscapes dataset.

This paper is not novel. Generally, it just utilized additional sources to help the generation. It is difficult to generalize to another dataset/domain as the CG-based set is not easy to get. The model side has very limited novelty. The pipeline and motivation are similar to previous work for domain adaptation for generative models and bridging domains:
1. Unsupervised Image-to-Image Translation Networks
2. Adaptation Across Extreme Variations using Unlabeled Domain Bridges

The experimental study is weak. Only the Cityscapes dataset is included. It is good to see if this method can work on others, like ADE20k and COCO-stuff. Also, some strong baseline, such as SPADE, is ignored.

**Experience Assessment:**

I have published one or two papers in this area.

**Review Assessment: Checking Correctness Of Derivations And Theory:**

N/A

**Review Assessment: Checking Correctness Of Experiments:**

I carefully checked the experiments.

**Review Assessment: Thoroughness In Paper Reading:**

N/A

---

### Official Review · AnonReviewer2 · 2019-10-23
**Official Blind Review #2**

**Rating:** 3

**Review:**

Authors use edge maps to generate more realistic images and videos in GANs. They train a GAN in a conditional way to
generate a photo-realistic version of a given scene.

In my opinion, the novelty of the proposed approach is not high and the improvements are incremental. I have some concerns about the method as well: what happens if the quality of edge maps is not good for some datasets? Is it realistic to assume that the edge maps of real images are provided with high quality?

Regarding the video generation, authors should compare their approach with recent video GAN papers such as https://arxiv.org/abs/1907.06571

There are several typos in the paper as well. I suggest that authors proof read their paper more carefully.

**Experience Assessment:**

I have published in this field for several years.

**Review Assessment: Checking Correctness Of Derivations And Theory:**

I assessed the sensibility of the derivations and theory.

**Review Assessment: Checking Correctness Of Experiments:**

I assessed the sensibility of the experiments.

**Review Assessment: Thoroughness In Paper Reading:**

I made a quick assessment of this paper.

---

### Official Review · AnonReviewer3 · 2019-10-31
**Official Blind Review #3**

**Rating:** 1

**Review:**

SUMMARY: Use HED to convert semantic maps to edge maps, then use GAN to generate realistic images, by adding several losses together. Also add a flow loss while generating videos.

Paper could be written much better, it is slightly confusing to follow.

The effort behind the work is commendable, they used good baselines to compare. However, the idea does not seem different from already existing ones. Although the authors mention this as a CG2real problem, as far as I understand this is a semantic maps to image problem. Maybe I'm missing something.

The only innovation is to have a middle step of converting these semantic maps to edge maps, and then from edge maps to realistic images. This is not altogether a big change, I was hoping to see it goes in a good direction. As far as I understand, the DNED model the authors use is a slight change to the HED model the authors cite, in that they sample different weights to combine to form different outputs.

The results look good from the examples shown. However, there is a lot more work to be done. Since architecturally there is not much difference in the already existing models, maybe focus on more use cases of edge maps. What is the advantage of using edge maps as an intermediate step, as opposed to some other representation? How do we exploit this fact better? These are good questions to explore.

The video results look good, and this is one example of exploiting the fact that edge maps are an intermediate step. However, more work needs to be done to support claims of diversity.

Overall, it seems intuitive to throw in a lot of losses and let the network optimize over all of them in an adversarial way, but a lot more effort needs to be put into thinking about any innovation there.

**Experience Assessment:**

I have published one or two papers in this area.

**Review Assessment: Checking Correctness Of Derivations And Theory:**

N/A

**Review Assessment: Checking Correctness Of Experiments:**

I carefully checked the experiments.

**Review Assessment: Thoroughness In Paper Reading:**

I read the paper thoroughly.

---

### Decision · Program_Chairs · 2019-12-19

**Decision:**

Reject

**Comment:**

The submission proposes a new GAN-based method for translating from semantic maps of (synthetic) images/videos (from computer graphics) to photo-realistic images/videos with the aid of edge maps. The main innovation is the inclusion of edge maps to the generator, where the edge maps are initially computed using the spatial Laplacian operator, and later output from their DNED network. According to the authors, the edge map allows them to generate images with fine details and to generate output images at higher resolutions.  The authors use their method to generate both single images as well as videos.

The submission received relatively low scores (2 rejects and 1 weak reject).  This was unchanged after the rebuttal (the authors did not submit a revised version of their paper).  The reviewers voiced concerns about the following:
1. Limited novelty
All of the reviewers indicated that they felt the novelty of the proposed approach of not high as the work seemed to make only small modifications on prior work.  In the author response, the authors provided some details on where they felt their innovation to be.  The paper can be improved by building on those and having experiments/examples to probe those claims in more detail.

2. Application to other datasets
The proposed method is demonstrated only on two datasets of driving scenarios (Cityscapes and Synthia).  It is unclear how the method will generalize to other types of inputs.  Experiments on other datasets will demonstrate whether the proposed approach can work well for other types of images.

3. The overall quality of the writing.
The overall quality of the writing is poor and hard to follow in places. The paper should also include more discussion of domain adaptation in the related work section.  It's possible that with improved writing that situates the work and explains the novel aspects of the work better, that the concern about limited novelty will be partially alleviated.   The paper also needs an editing pass as there are many grammar/spelling/capitalization issues.

Page 2: "We make Three" --> "We make three"
Page 3: "as can bee seen in fig 3.2" --> "as can be seen in ..." (it's unclear which figure "fig 3.2" refers to, as figures are labeled Figure 1, Figure 2, etc)
Page 4: Equation (3), symbol e is not explained (it is presumably the edge map)
Page 7: "bellow" --> "below"

Overall, there are interesting elements in this paper and the reviewers noted that the generated results look good.  However, the paper will need to be improved considerably.   The authors are encouraged to improve their work and submit to an appropriate venue.